# Comparative Evaluation of Recombinant and Acellular Pertussis Vaccines in a Murine Model

**DOI:** 10.3390/vaccines12010108

**Published:** 2024-01-22

**Authors:** Kyu-Ri Kang, Ji-Ahn Kim, Gyu-Won Cho, Han-Ul Kang, Hyun-Mi Kang, Jin-Han Kang, Baik-Lin Seong, Soo-Young Lee

**Affiliations:** 1The Vaccine Bio Research Institute, Annex to Seoul Saint Mary Hospital, College of Medicine, The Catholic University of Korea, Seoul 06591, Republic of Koreakjhan@catholic.ac.kr (J.-H.K.); 2The Interdisciplinary Graduate Program in Integrative Biotechnology, Yonsei University, Incheon 21983, Republic of Korea; 3Department of Pediatrics, College of Medicine, The Catholic University of Korea, Seoul 06591, Republic of Korea; 4Department of Microbiology and Immunology, College of Medicine, Yonsei University, Seoul 03722, Republic of Korea; 5Department of Pediatrics, Bucheon St. Mary’s Hospital, The Catholic University of Korea, Bucheon 14647, Republic of Korea

**Keywords:** efficacy of acellular pertussis vaccine, mouse study, DTaP vaccine, recombinant aP vaccine, PTx neutralization

## Abstract

Since the 2000s, sporadic outbreaks of whooping cough have been reported in advanced countries, where the acellular pertussis vaccination rate is relatively high, and in developing countries. Small-scale whooping cough has also continued in many countries, due in part to the waning of immune protection after childhood vaccination, necessitating the development of an improved pertussis vaccine and vaccination program. Currently, two different production platforms are being actively pursued in Korea; one is based on the aP (acellular pertussis) vaccine purified from *B. pertussis* containing pertussis toxoid (PT), filamentous hemagglutin (FHA) and pertactin (PRN), and the other is based on the recombinant aP (raP), containing genetically detoxified pertussis toxin ADP-ribosyltransferase subunit 1 (PtxS1), FHA, and PRN domain, expressed and purified from recombinant *E. coli*. aP components were further combined with diphtheria and tetanus vaccine components as a prototype DTaP vaccine by GC Pharma (GC DTaP vaccine). We evaluated and compared the immunogenicity and the protective efficacy of aP and raP vaccines in an experimental murine challenge model: humoral immunity in serum, IgA secretion in nasal lavage, bacterial clearance after challenge, PTx (pertussis toxin) CHO cell neutralization titer, cytokine secretion in spleen single cell, and tissue resident memory CD4+ T cell (CD4+ T_RM_ cell) in lung tissues. In humoral immunogenicity, GC DTaP vaccines showed high titers for PT and PRN and showed similar patterns in nasal lavage and IL-5 cytokine secretions. The GC DTaP vaccine and the control vaccine showed equivalent results in bacterial clearance after challenge, PTx CHO cell neutralization assay, and CD4+ T_RM_ cell. In contrast, the recombinant raP vaccine exhibited strong antibody responses for FHA and PRN, albeit with low antibody level of PT and low titer in PTx CHO neutralization assay, as compared to control and GC DTaP vaccines. The raP vaccine provided a sterile lung bacterial clearance comparable to a commercial control vaccine after the experimental challenge in murine model. Moreover, raP exhibited a strong cytokine response and CD4+ T_RM_ cell in lung tissue, comparable or superior to the experimental and commercial DTaP vaccinated groups. Contingent on improving the biophysical stability and humoral response to PT, the raP vaccine warrants further development as an effective alternative to aP vaccines for the control of a pertussis outbreak.

## 1. Introduction

Pertussis (whooping cough) is an upper respiratory disease caused by *Bordetella pertussis*. It is a highly contagious, infectious disease that causes coughing spells and can lead to complications and death [1,2]. In 1954, the diphtheria, tetanus, and whole-cell pertussis (DTwP) vaccine was introduced and used until the early 1980s, but due to the safety problems associated with the wP vaccine, an acellular pertussis (aP) vaccine containing pertussis toxoid (PT) and filamentous hemagglutinin (FHA) was developed [3]. In the late 1990s, a trivalent vaccine containing PT, FHA, and pertactin (PRN) was introduced and has been used globally for over 30 years [4]. Since the 2000s, occasional pertussis outbreaks have occurred in advanced countries, such as Europe, Australia, the United States, and Japan, mostly among youths and adults [5,6,7,8]. In 2015, the U.S. National Institute of Allergy and Infectious Diseases (NIAID) identified pertussis as an emerging infectious pathogen/disease, emphasizing the importance of novel vaccine development [9]. Also, intermittent small-scale pertussis outbreaks have gradually increased in Korea since the 2000s [10].

These outbreaks are probably due to the waning of DTaP and Tdap vaccines’ protective antibody titer against pertussis after vaccination [11,12,13,14], leading to a high infection rate in adolescents and adults. It is estimated that protection after natural infection lasts for 20 years, whereas vaccination provides relatively shorter protection; the wP vaccine for 12 years, and the aP vaccine for about 4 years [15]. Additional explanations of the outbreak are a lack of vaccine-induced protection against pertactin (prn)-deficient pertussis variants or strains with mutated ptxP alleles [16,17,18,19], and transmission from asymptomatic carriers [20]. In order to respond to these epidemiological changes, Tdap vaccination for adolescents and adults is actively recommended [21,22]. Preferably, a new vaccine that can respond to immunogenic waning effects and mutant strains is needed. Since all DTaP and Tdap vaccines in Korea are imported, domestic vaccine development is also crucial to ensure vaccine self-sufficiency in Korea.

In the 1980s and 1990s, when aP vaccines were developed, primary attention was given to humoral response [21,22,23,24], but later, Th1/Th17 cell-mediated immunity has been proven to play a more significant role for pertussis prevention [25,26,27,28]. Of note, lung tissue-resident memory T (T_RM_) cells are important, and IFN-gamma and IL-17a production induced by CD4+ T_RM_ cells are crucial in controlling pertussis infection [29,30,31]. Despite conflicting results, secretory IgA from the mucosal surface of the respiratory tract, mediated by IL-17a, reduces pertussis colonization and promotes long-term immunity [32,33]. Recent technical innovations on protein folding platforms are being tested for recombinant vaccines against a variety of infectious diseases [34,35,36,37,38]. This prompted us to develop and evaluate recombinant pertussis vaccine antigens from bacterial hosts as a potential alternative to previously developed aP vaccines.

In this study, two different types of vaccines, *B. pertussis* cell derived aP and recombinant raP, were compared with a commercially available vaccine from GSK. We evaluated the humoral and cellular immunogenicity, including lung tissue-resident memory CD4+ T cells (CD4+ T_RM_) and secretory IgA, of a trivalent DTaP vaccine containing three acellular pertussis antigens, PT, FHA, and PRN (developed by GC Pharma), and recombinant acellular pertussis (raP) vaccine (developed by Yonsei University and Vaccine Innovative Technology ALliance (VITAL), Korea), which was produced from *E. coli*. This study aims to evaluate the immunogenicity and protective efficacy in the experimental challenge model as an initial step toward developing a recombinant vaccine modality as an alternative to aP vaccines for the control of recurrent pertussis infection.

## 2. Methods

### 2.1. Vaccines

Figure 1 shows (a) the overall experimental scheme for mouse immunization and challenge, (b) vaccine groups, (c) and immunological assay systems after vaccination. The mock-vaccinated group (negative control) was inoculated with 0.85% physiological saline, while the positive control group received the Infanrix^TM^ IPV/Hib vaccine (GSK, Rixensart, Belgium). Infanrix^TM^ IPV/Hib vaccine (0.5 mL per dose) is composed of 25 Lf diphtheria toxoid (DT), 10 Lf tetanus toxoid (TT), 25 μg PT, 25 μg FHA, and 8 μg PRN adsorbed to aluminum hydroxide. The GC DTaP vaccine is a trivalent DTaP vaccine, which has the same quantity of each antigen as the Infanrix^TM^ IPV/Hib vaccine, and all antigens are adsorbed by aluminum hydroxide. GC DTaP vaccine’s PT, FHA, and PRN antigens are isolated from *B. pertussis* culture and combined with DT (Park-Williams #8 strain) and TT (Harvard strain), originated from GC Pharma (Green Cross Corp., Yongin, Republic of Korea)’s TD vaccine-prefilled Syringe Inj.^®^. The recombinant DTaP vaccine consists of GC Pharma’s diphtheria and tetanus toxoids (same concentration of GC DTaP), and three components of raP vaccine antigens produced by a WHEP vector (Figure 2a), developed at Yonsei University. WHEP domain derived from human glutamyl prolyl tRNA synthetase (hEPRS) was chosen as a fusion partner for exerting chaperna function [36,37,38] to expedite folding and soluble expression in *E. coli*. Thus, all three antigens-pertussis toxin subunit 1 (PtxS1), filamentous hemagglutinin (FHA), and pertactin (PRN) domains, respectively, were genetically fused with WHEP domain and expressed as soluble form, purified by Ni-affinity chromatography following similar protocol for purification (Figure 2b) [35]. FHA is an exceptionally large protein with the size of 220 kDa and a truncated domain ‘Mal85′ (1655-2111aa of FHA), previously identified as crucial for its immunogenicity, was used [39]. Additionally, the endotoxin of purified antigens was removed by phase separation method using Triton X-114 (Sigma-Aldrich, Saint Louis, MO, USA) at 1%. The trivalent recombinant acellular pertussis antigens and GC Pharma’s diphtheria, and tetanus solution were adsorbed to 1mg/mL of Alhydrogel^®^ adjuvant (Invivogen, San Diego, CA, USA) separately, and mixed to have a concentration of GC DTaP.

### 2.2. Mouse Vaccination

Four-week-old female BALB/c mice (Orient Bio, Seongnam, Republic of Korea) were divided into four groups: a mock-vaccinated (negative control) group, a commercial vaccine (positive control) group, and test groups 1 and 2 (Figure 1b). The mice were inoculated intraperitoneally three times at a dose of 0.125 mL, which corresponds to 1/4 of the human dose (0.5 mL). The mice were housed in filter-top cages under semi-specific pathogen-free conditions, and food and water were freely available. All animal research procedures were performed in accordance with the Laboratory Animals Welfare Act, the Guide for the Care and Use of Laboratory Animals, and the Guidelines and Policies for Rodent Experiments provided by the IACUC (Institutional Animal Care and Use Committee) of the School of Medicine, The Catholic University of Korea (approval number: CUMS-2021-0054-04). The animals were vaccinated three times at 2 or 3 week intervals (Figure 2). Blood was collected 6 times, 3 weeks after the second dose (5 weeks), 3 weeks after the third dose (8 weeks), and 1, 2, 4, and 6 weeks post infection (p.i.), and collected from 5–8 animals per group at each time point. Lung, nasal lavage fluid, and spleen samples were collected pre-infection (8 weeks), 2 h post-infection (p.i.), at day 5 p.i., 1 week p.i., 2 weeks p.i., and 6 weeks p.i. (Figure 1a) to investigate cellular immunity, bacterial clearance, and intranasal IgA levels. Samples were collected after anesthetizing the mice using intraperitoneal Zoletil 50^®^ Inj.(Virbac, Carros, France) and Rompun^®^ Inj. (Bayer, Leverkusen, Germany), while the nasal lavage fluid was collected using a 24 gage catheter, by flushing the nose with 200 uL of saline.

### 2.3. ELISA Assay

The serum IgG, IgG1, and IgG2a titers against the three pertussis antigens, PT, FHA, and PRN, were evaluated using an indirect enzyme-linked immunosorbent assay (ELISA). The assays were conducted according to previous methods [40], with modifications. Immunoplates were coated with 0.1, 0.15, and 0.25 μg/mL of PT, FHA, and PRN antigens obtained from NIBSC (National Institute for Biological Standards and Control, South Mimms, UK), and the plate was incubated overnight. Mouse International reference serum (NIBSC 97/642) [41] and the samples were added to the wells and incubated for 2 h at room temperature. After washing the plates, HRP-conjugated secondary anti-mouse IgG, IgG1, or IgG2a antibodies were added and incubated for 1 h. After reacting with the substrate TMB, 100 μL of the stopping solution 0.16 M H_2_SO_4_ was added to each well, and the OD_450_ was measured. For all experiments, Japanese anti-FHA mouse serum (NIBSC, JNIH-11), anti-PT mouse serum (NIBSC, JNIH-12), and NIBSC 97/642 were used as quality controls, and the method was validated. The results were obtained using the reference line method. For the IgA assay, the nasal lavage fluid was diluted with PBS in a 1:1 ratio and added to each well. After incubating with the IgA secondary antibody, the same procedure was used to measure the OD_450_. The diphtheria and tetanus humoral immunogenicity results were measured using an ELISA kit from ADI (Alpha Diagnostic International Inc., San Antonio, TX, USA)The diphtheria results were calculated as an index by dividing the sample OD by the 10 U/mL calibrator OD. Values greater than 1.0 were taken as positive, and values < 1.0 as negative. For tetanus, results were calculated using an STD curve using the point-to-point method to interpolate between calibrators at four different concentrations. All samples for diphtheria and tetanus humoral immunity were diluted in a ratio of 1:20,000 before testing.

### 2.4. PTx CHO Cell Neutralization Assay

This assay was performed according to previous methods, to analyze the level of neutralizing antibodies to pertussis toxin (PTx) in Chinese hamster ovary (CHO) cells (ATCC, CCL-61) [42,43]. The CHO cells were incubated for at least 2 days in Ham’s F-12K (Kaighn’s) Medium with 10% fetal bovine serum (FBS) before adding 100 μL to each well of a 96 well cell culture plate at a concentration of 1.5 × 10^5^/mL. The serum was diluted with F-12K medium in a two-step series (1:8–1:16,384) and PTx (NIBSC 15/126, 1-week storage) was prepared with 4 CPU (0.27 IU/mL, 7.81 ng/mL) concentration. 50 μL of each serum and 4 CPU of PTx was mixed and incubated at 37 °C for minimum 2 h. Then, 100 μL of the reacted serum/PTx mixture was added to the prepared CHO cells and incubated in a 5% CO_2_ incubator at 37 °C for 24 h. After removing the medium and washing it with DPBS, the cells were fixed with 10% formaldehyde solution for 3 min, stained with 0.5% Crystal Violet Solution, and evaluated by microscopy. If CHO cell clustering occurred in more than 10 cells, it was recorded as clustering positive. The serum titer was determined based on the serum dilution ratio at which CHO cell clustering started and the titer started with a 1:8 dilution. Samples with titers below 1:8 were attributed to 1:4 for statistical analysis.

### 2.5. Bacterial Clearance

For the bacterial challenge, we applied a previous method from our laboratory [44]. In accordance with the WHO annex 4, a standard pertussis strain (ATCC 9797) was suspended in 40 μL of 1% casamino acid solution, at a concentration of 5 × 10^6^ colony-forming units (CFUs) and administered intranasally to mice anesthetized with Zoletil 50^®^ Inj. and Rompun. Lung tissue was harvested 2 h, 5 days, 1 week, and 2 weeks post infection, homogenized using a microtube BeadBug™ homogenizer (Benchmark Scientific Inc., Sayreville, NJ, USA), and cultured on charcoal agar for more than 4 days to determine colony forming units (CFUs) and converted to Log10 to assess CFUs by time points.

### 2.6. Cytokine ELISA

Based on previous methods [44], we homogenized spleen tissue using a single cell dissociator (RWD, Sugar Land, TX, USA), filtered the homogenate through a 70 μM cell strainer, and then performed RBC lysis to prepare the cells to a concentration of 5 × 10^6^/mL. After that, 100 μL of each sample was added to a 96 well plate and cultured for 3 days before collecting the supernatant. Commercially available cytokine ELISA kits from Proteintech (Rosemont, IL, USA) for IL-17a, IL-5, and INF-gamma were used. Here, 1 × 10^6^ CFU/mL of hBp (heat-inactivated pertussis), 8 μg/mL of PT, 8 μg/mL of FHA, and 4 μg/mL of PRN were used as stimulators; 2 μg/mL LPS (Sigma-Aldrich, Saint Louis, MO, USA) was used as a positive control.

### 2.7. T Cell Surface Staining via Flow Cytometry

An amount of 2 μg of anti-mouse CD45-BV786 antibody (BD Biosciences, Franklin Lakes, NJ, USA) was diluted in sterile PBS and administered intravenously in the tail vein 10 min before anesthesia. CD 45+ is a circulating lymphocyte, and CD45− is a tissue-resident cell. In total, 40 U/mL DNase I and 1 mg/mL Collagenase IV enzyme were measured to a volume of 2 mL and incubated with lung tissue at 37 °C for 30 min. The tissue was then homogenized using a dissociator, filtered through a 70 μM cell strainer, and subjected to RBC lysis to obtain single lung cells. After preparing the lung cells to a concentration of 5 × 10^6^ cell/mL, they were washed with PBS and live-cell staining was performed using fixable viability stain 700 (FVS700) (BD Biosciences, Franklin Lakes, NJ, USA). After treating lung cells with Fc Block^TM^ reagent, the surface markers CD44-BV421, CD69-BV605, CD103-PE, CD62L-APC, CD4-FITC, CD3-PE-Cy7, and CD8-BB700-per cycle 5.5 ((BD Biosciences, Franklin Lakes, NJ, USA)) were used for staining. The flow cytometry gating strategy was performed according to Appendix A Appendix A, based on a previous study [45]. FACS Aria Fusion from BD Biosciences was used, and all results were derived using FlowJo software v10.

### 2.8. Statistical Analysis

All results are shown as mean ± standard errors of the mean (SEM) and were compared using one-way ANOVA with Bonferroni’s post hoc test. Statistical analysis was performed using GraphPad Prism^TM^ software v9 (GraphPad, San Diego, CA, USA), and statistical significance was defined as a *p* value (* *p* < 0.05, ** *p* < 0.01, *** *p* < 0.001, **** *p* < 0.0001).

## 3. Results

### 3.1. Humoral Immunity to PT, FHA, and PRN Antigens

Humoral immunity is mainly induced through the aP vaccine, and research on this vaccine was actively conducted in the 1980s and 1990s when its development first began [21,23,24]. A previous experiment with infant baboons [28] showed that the aP vaccine and humoral immunity play a relatively minor role as compared with cell-mediated immunity and protecting against infection. However, the aP vaccine is still important in certain groups, such as infants and young children who are at a high risk of exposure to pertussis [2], pregnant women in the final stages of pregnancy, and medical workers [46]. In particular, the fetus is protected from pertussis infection after birth by maternally transferred antibodies [46,47,48]. The type of T helper responses (Th1 and Th2) are correlated with the IgG isotypes [49,50]; Th1 response is correlated with IgG2a/IgG2c in mice [51,52] and IgG1 in humans, whereas Th2 is mainly associated with IgG1 in mice [53] and IgG4 in humans [54,55].

Overall, humoral immunity in serum showed high titers at 8 weeks, corresponding to 3 weeks after boost vaccination (Figure 3). Of note, the titer did not further increase after subsequent infection, suggesting that a maximal antibody response was already attained by vaccination (Figure 3). However, variable responses were observed among different groups; high response for PT by GC DTaP, FHA by control, PRN by GC DtaP (IgG and IgG1), and notably for PRN by raP (IgG2a) (Figure 3). The raP group showed unusually low titers of PT antigen and showed statistical differences in IgG and IgG1 for all vaccinated groups at all time points (Figure 3a). The GC DTaP group showed a statistically significant higher response than the control group for PT and PRN (Figure 3a,c), and a lower response for FHA before and after the challenge (Figure 3b).

For diphtheria, all groups except for the mock-vaccinated group showed a positive index of 1 or more (≥1.0). For tetanus, titers were similar among all groups after vaccination, and both GC and raP groups maintained relatively higher titer than the control group after the bacterial challenge. (Appendix A).

### 3.2. PTx CHO Cell Neutralization Assay

An opsonizing antibody is known to activate the complement system via complement receptor 3 (CR3) and the IgG Fc receptor, leading to the activation of cell-mediated immunity [22,23,47]. Opsonizing antibodies help macrophages and neutrophils to uptake or kill bacteria by binding to pertussis surface antigen [48,49]. We used a PTx CHO cell neutralization assay to measure opsonizing antibodies to PTx in serum at 3 weeks after boost vaccination (8w), and at 1, 2, and 6 weeks post-infection. Of note, the raP group, albeit with low humoral immunity to PT (Figure 3a), the CHO cell protection titer was as high as the other vaccinated groups 3 weeks after boost vaccination (Figure 4); although, the raP group showed a lower titer than the control and GC DTaP groups after infection (Figure 4). The mock-vaccinated group showed almost no titer until 6 weeks post-infection. Opsonizing antibodies against PTx persisted until 6 weeks post infection. There was no significant difference between GC DTaP and the control groups before and after infection (Figure 4).

### 3.3. Nasal Lavage IgA

Mucosal IgA plays an important role in intranasal immunity. Specifically, IgA can kill intranasal *B. pertussis* through opsonization or amplifying the IL-17a expression by T_RM_ cells, contributing to long-term immunity [28,33]. Natural infection or wP vaccination can mainly induce mucosal IgA, but its induction is low in the aP vaccine [28,56]. In accordance with the WHO annex 4, we challenged vaccinated mice intranasally with *B. pertussis* ATCC 9797 strains. We collected nasal lavage before infection and 2 h and 1, 2, and 6 weeks post infection, and measured the levels of IgA. Vaccination induced a significant increase in the IgA level (most notably raP group) as compared with MV group; although, the IgA level was relatively low for PT or PRN (OD value < 0.6), and there were no between-group differences (Figure 5a,c). FHA-specific IgA was much more pronounced (OD up to ~2) than PT and PRN and showed a steady increase post-infection. The control group showed the strongest response, at 2 weeks post infection, which differed significantly from the raP group (Figure 5b).

### 3.4. Bacterial Clearance

All vaccination groups exhibited statistically significant improvement of bacterial clearance over the entire period of observation post-infection, in clear contrast to the MV group, where bacterial titer was briefly increased 5d post infection due to prolific infection. The raP group showed slower clearance kinetics than the control group but provided almost complete clearance 2 w post-infection (log10 CFU < 0.4) (Figure 5d).

Opsonizing antibodies play an important role in bacterial clearance in the early stage of pertussis infection [57,58], and in the murine model, IgG2a and IgG2c isotypes are associated with opsonizing antibodies. It is presumed that the kinetics of bacterial clearance during the course of infection are an outcome of multiple factors, as reflected in the CHO protection opsonization titers (Figure 4), probably triggered by IgG2a isotypes (Figure 3) and on-site protection at the respiratory tract by secretory IgA antibodies (Figure 5). Reportedly, PT has a considerable effect on the initial stage of pertussis infection [59,60].

However, bacterial clearance is affected not only by the opsonizing antibodies but also, and more importantly, the CD4 Th1- and Th17-mediated adaptive immune system [58,61,62]. The raP group showed almost as sterile protection as other vaccinated groups at 2 weeks after infection (Figure 5d), despite lower anti-PT antibody level (Figure 3a). Therefore, we further examined the potential contribution of Th1 and Th17 to protective immune responses.

### 3.5. Cytokine Response

The expression levels of IFN-gamma and IL-17a upon heat-inactivated pertussis (hBp) stimulation were much higher than those upon stimulation with individual single component (compare <1000 units for PT (Appendix A), FHA (Appendix A), and PRN (Appendix A with up to 6000 units for hBp), (Appendix A Appendix A). IFN-gamma and IL-17a are mainly expressed through natural infection or wP vaccination, but poorly expressed by aP vaccination [27,28,63], and long term protection from the wP vaccination could be explained with it.Assuming that stimulation with heat hBp mimics the response to natural infection [44], the highest responses of IFN-gamma and IL-17a to hBp stimulation are consistent with the results of a previous study conducted in our laboratory [44]. Consistently, 2 weeks post-infection, the mock-vaccinated group showed equal or superior results to other groups, in terms of IFN-gamma and IL-17a (Figure 6a,b, Appendix A Appendix A). Most notably, the raP group, among all groups tested, including the commercial control group, showed the highest-level stimulation for both IFN-gamma and IL-17a at 1 week post-infection. Unlike IFN-gamma and IL-17a, IL-5 showed similar expression levels in response to hBp and each of the three antigens; although, the expression level was variable among individual components (PT, FHA and PRN) before and after challenge (Figure 6c–f). The distinctive stimulation of IFN-gamma and IL-17a may contribute to the sterile protection offered by the raP vaccine (Figure 5d).

### 3.6. Tissue-Resident Memory T Cell

When we analyzed CD4+ T_RM_ cell expression, the overall CD4+ T cell ratio increased in all groups 2 weeks post-challenge and decreased to pre-challenge levels after 4 weeks (Figure 7). The CD4+ T_RM_ cell ratio increased post-challenge compared to pre-challenge. In particular, at 2 and 4 weeks post-challenge, the mock-vaccinated group showed a large increase in CD4+ T_RM_ cell expression, exhibiting significant differences from all the other groups (Figure 7b). Notably, the raP group showed higher CD4+ T_RM_ cell expression at 2 weeks post challenge than the control and GC groups (Figure 7b). Except for this, no clear differences were observed in GC or positive control groups.

## 4. Discussion

Small-scale pertussis outbreaks have gradually increased in many countries, including Korea [10], probably due to the waning of DTaP and Tdap vaccines’ protective efficacy against pertussis [11,12,13,14]. As a re-emerging infectious pathogen/disease, pertussis requires the strengthening of a vaccination program as well as novel vaccine development [9]. A growing need for new vaccine modality, we evaluated the recombinant raP antigens produced from *E. coli* compared with aP antigens purified from *B. pertussis* in this study. To overcome misfolding into immunologically irrelevant inclusion bodies in *E. coli* host, a novel chaperna approach was undertaken to increase the proper folding and the solubility of recombinant antigens [36,37,38]. aP components were combined with diphtheria (D) and tetanus (T) toxoids into DTaP trivalent vaccine, and the immunogenicity and the protective efficacy of raP and aP were compared in an experimental murine immunization and challenge model using Infanrix^TM^ as a comparator vaccine (control). Various criteria for analysis included humoral immunity in serum, mucosal IgA secretion in nasal lavage, PTx CHO cell neutralization titer, cytokine secretion in isolated spleen cells, and tissue resident memory CD4+ T cell in lung tissues, and finally, by bacterial clearance after challenge.

In humoral immunogenicity, the contribution of individual antigen for each group was clearly identified, albeit with a degree of variability among three components, PT, PRN, and FHA (Figure 3). In short, among three different preparations, the best performance was noted in PT in GC, FHA in control, and PRN in raP (especially in IgG2a isotype). Time-dependent antibody titer level was similar to our Tdap booster previous study [44], showing the highest antibody titer at 8 w, 3 weeks after the third vaccination, and gradually decreased thereafter. Notably, very low response in PT was observed in raP. This may explain the relatively low PTx CHO cell neutralization by raP (Figure 4). However, raP exhibited similar kinetics of bacterial clearance after bacterial challenge: a little delayed as compared with other groups, but eventually into the sterile clearance of lung infectious titer in 2 w p.i. In mice, IgG2a or IgG2c isotypes of humoral immunity is responsible for the opsonization of infected cells into bacterial clearance [57,58]. The results could be ascribed to a high level of mouse IgG2a or IgG2c related to opsonizing antibodies important for initial bacterial clearance [57,58]. Of note, raP exhibited relatively high level of IgG2a in anti-PRN antibody (Figure 3c) and may be crucially involved in sterile protection from bacterial challenge.

Memory T cell including CD4+ T_RM_ cell immunity is mediated by Th1 and Th17. CD4+ T_RM_ cells in the lungs are polyclonal [64] and play an important role in long-term immunity in cases of re-infection [31,65,66]. CD4+ T_RM_ cells are potently induced through natural infection or wP vaccination, but relatively weakly by aP vaccination [67,68,69]. The present study is consistent with this. First, the mock-vaccination (MV) group, akin to natural infection, showed high CD4+ T_RM_ cell expression and a significant difference from the other groups after bacterial challenge (Figure 7). The characteristics of natural infection were confirmed not only in terms of CD4+ T_RM_ cell response, but also in terms of cytokine response; IFN-gamma and IL-17a responses in MV group were equal or superior to the control and DTaP vaccinated groups after challenge (Appendix A Appendix A). It should be noted that, in the case of the recombinant raP vaccine, the expression rate of CD4+ T_RM_ cells was relatively higher than the other test groups (Figure 7b) at 2 w p.i. In addition, the IFN-gamma and IL-17a cytokine response after hBp stimulation from raP group was also superior to other vaccination groups at 1 w p.i. This may explain why the raP group showed sterile clearance of lung bacterial titer comparable to other groups. The results obtained for the raP-vaccinated group in this study are consistent with those previously reported by Fry et al. [70]. Using a genetically inactivated PtxS1 DNA vaccine, they showed that, despite very low IgG response in the serum, the response to splenocyte IFN-gamma and IL-2 was high and bacterial clearance was higher and faster than in the DTaP vaccine group. In the present study with raP vaccine, the bacterial clearance was manifest even with a very low PT antibody, probably reflecting significant role of T cell response and associated IFN-gamma and IL-17a cytokines induced by the raP vaccination. As a new modality of the pertussis vaccine, based on recombinant production from *E. coli,* the present report nonetheless requires further consideration in future.

Through the research on the live aP vaccine BPZE1 and the outer-membrane vesicle pertussis vaccine, secretory IgA (SIgA) and IL-17a, which induce eradication of mucosal aP bacteria, are triggered only through nasal administration [33,71]. In the present study, all vaccines were administered IP, and the expression of IgA in the nasal cavity was evident only after challenge. It was FHA that exhibited the highest value in IgA in nasal lavage, in line with a previous report on FHA-specific nasal IgA immunity via SC administration [72]. The MV group showed particularly high IL-17a secretion from FHA antigen stimulation in spleen cells (Appendix A). These results underscore the potential role of FHA for IgA and IL-17a. Otherwise, relative contribution among the three antigen components (Figure 5) and the correlation between IgA and bacterial clearance of nasal tissues was not confirmed.

The recombinant aP antigens were refractory to soluble expression in *E. coli*). In the present study, soluble production of raP antigens was ensured by fusion to an RNA-interaction domain (in this case, WHEP domain [39]) to harness with the chaperna function exploiting a novel chaperone function of tRNAs [34,38]. During the course of the study, we noted, despite initial soluble expression, an aggregation tendency in particular for recombinant PT_X_ during prolonged storage. This may be responsible for its low immunogenicity (Figure 3a). Therefore, the biophysical characteristics of these fusion proteins, e.g., solubility and aggregation behavior, need to be further monitored and improved to ensure long-term stability.

Currently licensed aP vaccines may have limitation to pertussis variations, and it is likely that recombinant vaccines could provide an option for rapid response to sporadic pertussis outbreak by genetic variants. For instance, vaccine-derived immunity may not protect against mutant pertussis strains, particularly against pertactin (prn)-deficient variants [16,17,18,19]. In the 1990s, pertussis epidemics returned to advanced nations despite high vaccination rates, such as the Netherlands, the United States, Japan, the UK, and Austria [72]. There have been claims of PTx differentiation globally (e.g., ptxP3 allele replacing ptxP1) by pathogenic adaptation [73]. The prevalence of pertussis has also increased in Korea since 2009, due to the emergence of the ptxP3 allele [74]. Similarly, PRN-deficient *B. pertussis* also occurs in advanced countries and cause adaptations to aP-vaccinated humans [75]. The current licensed wP and aP vaccines were found less effective against PRN-deficient isolate than a PRN (+) isolate [76]. The Tohama I strain, which is usually used in commercial production of aP vaccines, has the ptxP1-ptxA2, prn1, fim2-1, and fim3-1 genotypes, and may have limited cross-protection against genetic variants. Recombinant production, as initiated in the present study, would provide an alternative, and hopefully, a better option for rapid response to re-emerging pertussis infection.

## 5. Conclusions

GC DTaP had equivalent efficacy to the licensed control group and warrants early phase human clinical study. The recombinant aP vaccine exhibits a slightly different profile in immunological parameters but show sterile protection from bacterial challenge. raP show superior cellular responses-CD4+ T_RM_ cell expression and IFN-gamma and IL-17a secretion-, probably compensating for a low PT_X_ humoral antibody and CHO neutralization titer toward protection. Contingent on improvement on humoral immunity to PT_X_, raP may provide a new modality for pertussis vaccine.

## Figures and Tables

**Figure 1 vaccines-12-00108-f001:**
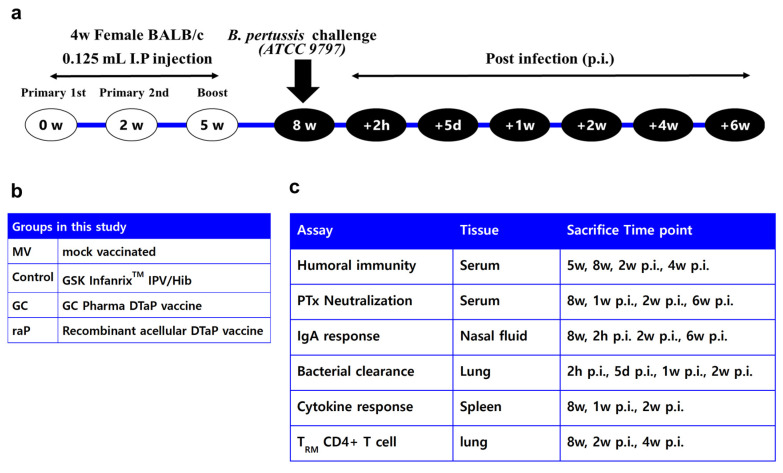
Scheme of the study. Four-week female BABL/c mice were vaccinated according to (**a**) schedule of the study and (**b**) groups in this study as shown. All assays performed at each time point indicated in (**c**) assay lists with 5–8 mice per group.

**Figure 2 vaccines-12-00108-f002:**
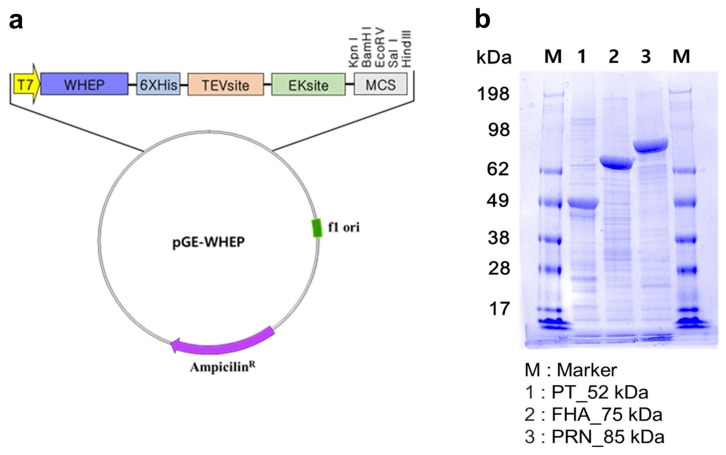
Diagram of WHEP vector and SDS-PAGE results of recombinant pertussis antigens. Recombinant pertussis antigens were produced by (**a**) WHEP vector. WHEP is a chaperna domain of human origin, derived from glutamyl prolyl tRNA synthetase. Hexa-histidine tag inserted to the vector for Ni-affinity purification. Purified antigens were indicated by (**b**) SDS-PAGE analysis of raPs. TEV, site-specific TEV protease recognition sequence; EK, site-specific enterokinase recognition sequence; MCS, multi-cloning site.

**Figure 3 vaccines-12-00108-f003:**
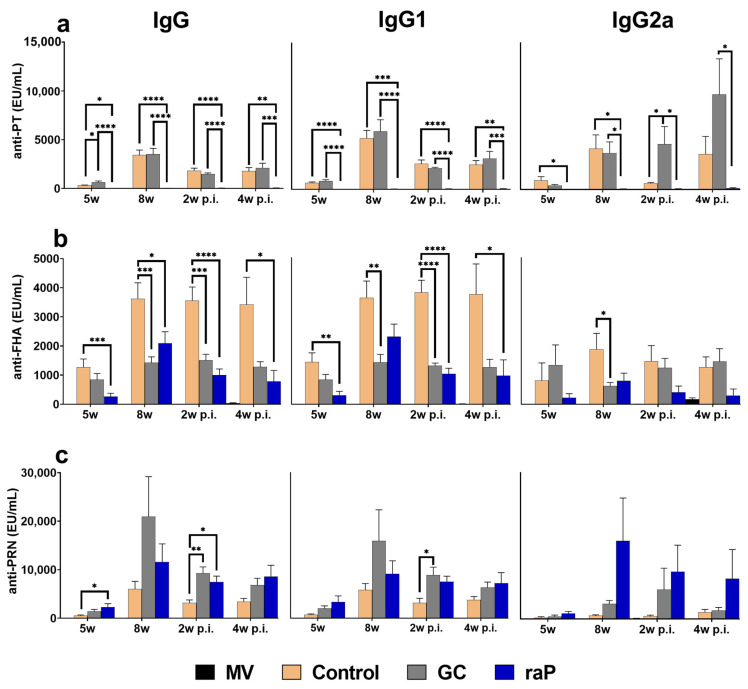
Humoral responses. IgG, IgG1, and IgG2a antibody titers to vaccine antigens (PT NIBSC 15/126, FHA NIBSC JNIH-4, PRN NIBSC 18/154) were determined using ELISA assay. Serum was collected and analyzed at 3 weeks after inoculation with the second primary vaccine (5 weeks), 3 weeks after inoculation with the booster (8 weeks), and at 2 and 4 weeks post-infection with pertussis (ATCC 9797). Data are presented as mean ± standard errors of the mean (SEM) (EU/mL). Results were obtained from eight animals per group at 5 weeks and 8 weeks, and five animals per group post-infection. (**a**) PT. (**b**) FHA. (**c**) PRN. Between-group differences were investigated using one-way ANOVA and Bonferroni’s post hoc test, and *p* values are visualized as follows: * *p* < 0.05, ** *p* < 0.01, *** *p* < 0.001, **** *p* < 0.0001. The differences with the mocked-vaccine group are not shown. EU, ELISA unit; MV, mock vaccinated.

**Figure 4 vaccines-12-00108-f004:**
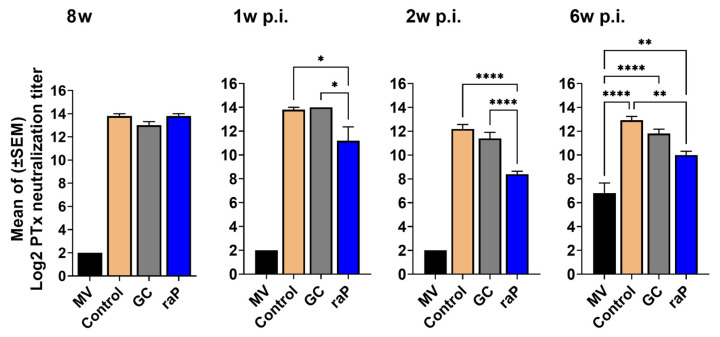
PTx CHO cell neutralization assay. Serum was collected at 3 weeks after the booster injection (8w) and at 1, 2, and 6 weeks post infection (p.i.) with *B. pertussis* (ATCC 9797), was serially diluted two-fold, then was reacted with NIBSC 15/126 PTx at a concentration corresponding to 4 CPU, and then cultivated with CHO-K1 cells for 24 h. The serum titer was measured as the dilution ratio at which the CHO cell clustering began. All results were transformed to Log2, then expressed as mean ± standard errors of the mean (SEM). Serum was collected from five mice per group. Between-group differences were investigated using one-way ANOVA and Bonferroni’s post hoc test, and *p* values are displayed as follows: * *p* < 0.05, ** *p* < 0.01, **** *p* < 0.0001. Differences with the mock- vaccinated group only presented at 6 w p.i., since differences of **** *p* < 0.0001 were observed at 8 w, 1 w p.i., and 2 w p.i. MV, mock vaccinated.

**Figure 5 vaccines-12-00108-f005:**
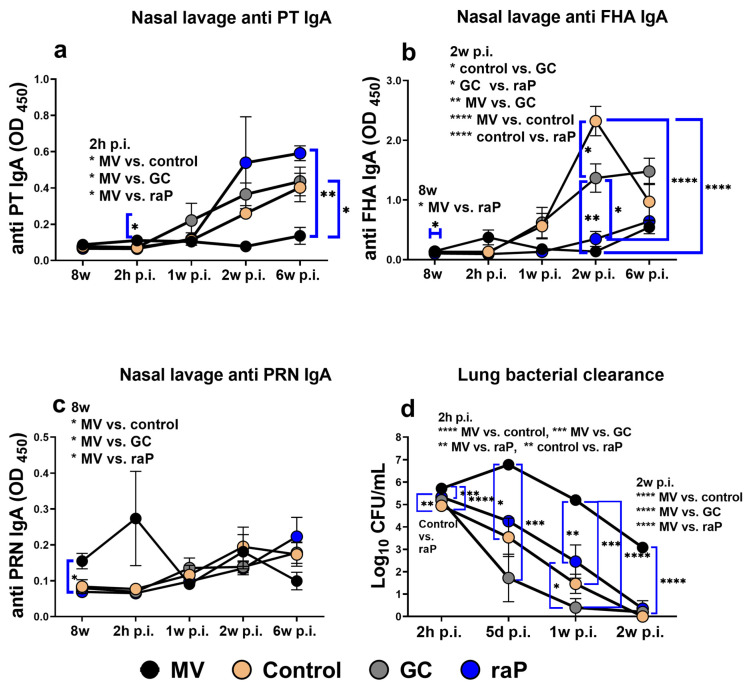
IgA of nasal lavage fluid and bacterial clearance of lungs. After three rounds of vaccination, we measured IgA levels in nasal lavage fluid (1:1 diluted with PBS) for each antigen at 8 w and at 2 h and 1, 2, and 6 weeks post infection (p.i.). All results are presented as the mean ± standard errors of the mean (SEM) of the OD_450_ values for five animals per group. We also investigated the levels of bacteria (CFU) in lung tissue from mice challenged with the same method at 2 h, 5 days, and 1 and 2 weeks p.i. (**a**) PT., (**b**) FHA., (**c**) PRN., (**d**) Lung homogenate was serially diluted ten-fold with casamino acid solution and cultured in charcoal agar for at least 5 days in a CO_2_ incubator before measuring the CFU. Between-group differences were investigated using one-way ANOVA and Bonferroni’s post hoc test, and *p* values are displayed as follows: * *p* < 0.05, ** *p* < 0.01, *** *p* < 0.001, **** *p* < 0.0001. CFU, colony-forming unit; MV, mock vaccinated.

**Figure 6 vaccines-12-00108-f006:**
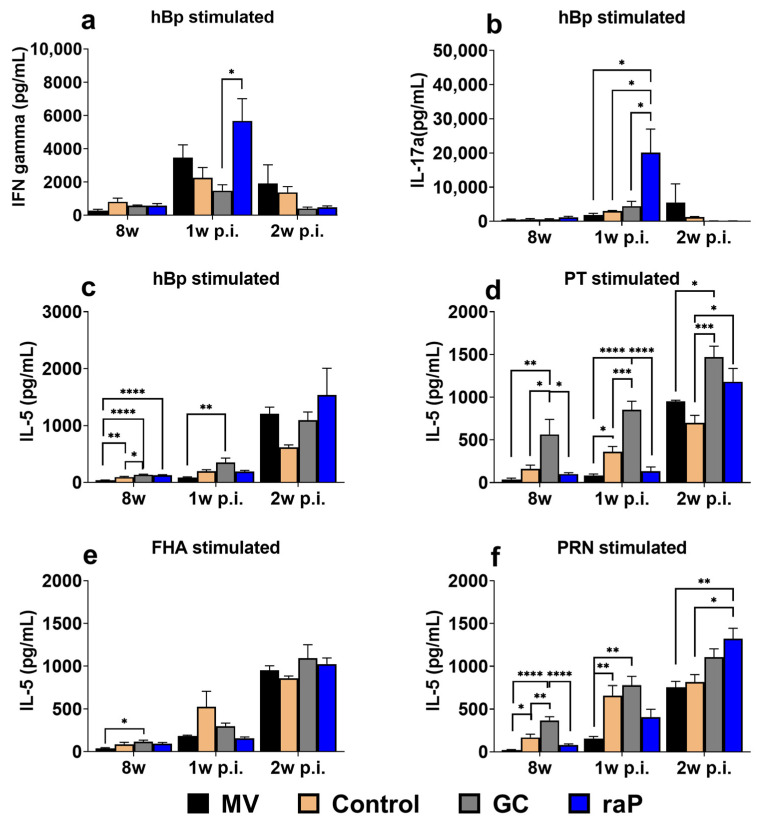
Cytokine response. Heat-inactivated pertussis (hBp) and individual solutions of the antigens PT, FHA, and PRN (obtained from GC and prepared to their vaccine concentrations of 25 μg/mL, 25 μg/mL, and 8 μg/mL, respectively) were cultured for 72 h with single splenocytes collected from the spleen at 8 weeks (3 weeks after boost vaccination) and 1 and 2 weeks post-infection (p.i.) with pertussis (ATCC 9797). After culture, the supernatant was collected, and cytokine ELISA kits (Proteintech) were used to measure the levels of IFN-gamma, IL-5, and IL-17a for each group (*n* = 5). To prepare heat-inactivated pertussis, the bacteria were adjusted using McFarland standards to a concentration of 1 × 10^6^ CFU/mL, suspended in pH 7.4 PBS, and heated at 65 °C for 30 min. (**a**) hBP-stimulated IFN-gamma secretion; (**b**) hBP-stimulated IL-17a secretion; (**c**) hBP-stimulated IL-5 secretion; (**d**) PT-stimulated IL-5 secretion; (**e**) FHA-stimulated IL-5 secretion; (**f**) results for PRN-stimulated IL-5 secretion. All results are presented as the means ± standard errors of the means (SEM). Statistical differences were investigated using one-way ANOVA and Bonferroni’s post hoc test. *p* values are as follows: * *p* < 0.05, ** *p* < 0.01, *** *p* < 0.001, **** *p* < 0.0001. MV, mock vaccinated.

**Figure 7 vaccines-12-00108-f007:**
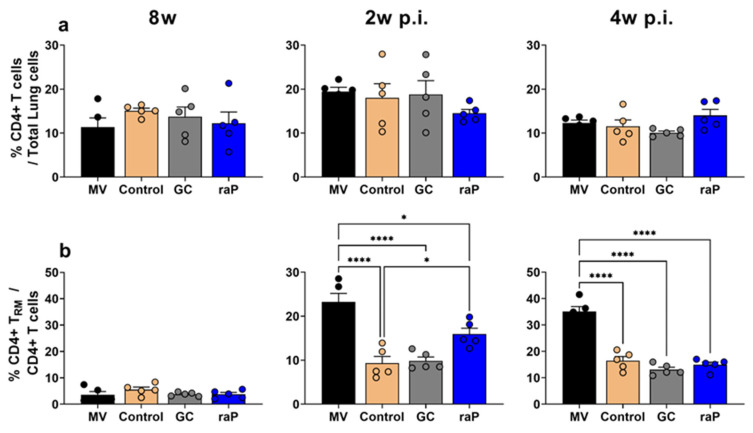
Expression of CD4+ T cells and CD4+ T_RM_ cells. CD4+ T cells and T _RM_ cells in lungs from five mice per each group were analyzed with FACS Aria Fusion at 8 weeks and 2 weeks post-infection (p.i.) and 4 weeks p.i. (**a**) Accumulation of CD4+ T cells among 10,000 lung cells are shown as a percentage. (**b**) The graph shows the percentage of CD4+ T _RM_ cells among total CD4+ T cells in lung tissue-resident cells (CD3+ CD45.2+ CD4+). All groups were compared using one-way ANOVA and Bonferroni’s post hoc test. *p* values are as follows: * *p* < 0.05, **** *p* < 0.0001.

## Data Availability

The datasets used and/or analyzed during the current study are available from the corresponding author upon reasonable request.

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
