# Peer review of "Comparative Evaluation of Recombinant and Acellular Pertussis Vaccines in a Murine Model"

_vaccines, 2024, doi:10.3390/vaccines12010108_

Round 1
Reviewer 1 Report
Comments and Suggestions for Authors
This manuscript evaluated two Pertussis vaccines, GC Pharma DTaP vaccine and recombinant acellular DTaP vaccine, and compared to GSK Infanrix IPV/Hib vaccine in mice. Comprehensive immunological monitoring, including IgG titers, lavage IgA titers, CHO cell protection, lung bacteria clearance, cytokine responses, and T cell responses.
Differential results among the three vaccines tested were observed. However, it not clear what caused these differential results and further implications for vaccine development.
The manuscript doesn't demonstrate enough significance in the results and
soundness in general.
Lots of assays were performed comparing these vaccines, without a clear
conclusion. In addition, in Figure 4 CHO cell protection assay, which is one
of the most important readout, the positive control vaccine from GSK
performed better. Thus, is unclear if the two vaccines tested are worth
development.
Additionally, many figures in this paper are hard to read, either not
properly labeled or visually hard to read.
Finally, a key point for this paper is to develop domestic Pertussis vaccine
for Korea. I don't think this would benefit the science community and global
public health.
The writing is understandable but need extensive editing. For instance, line 229-230 "IgG and IgG1 showed almost identical patterns, which is believed to be because both subtypes show a Th2 orientation" is not clear. There are many other writings like this in this paper.
Author Response
We would like to thank you for allowing us to revise the manuscript. We have revised the manuscript based on the comments. Additionally, we have made extensive revisions that we believe improved the quality of the manuscript. All authors have reviewed the revised version and approved it. Our point-by-point responses to the specific comments made by you and descriptions of additional changes are set out below.
Reviewer 1
- Differential results among the three vaccines tested were observed. However ,it is not clear what caused these differential results and further implications for vaccine development. The manuscript doesn’t demonstrate enough significance in the results and soundness in general. Lots of assays were performed comparing these vaccines without a clear conclusion.
Response : We appreciate your valuable comment. We discussed about the comment and agree with it. We had provided unclear conclusion. We revised the entire manuscript_ title, abstract, result, discussion, conclusion section for the clear conclusion.
- In Figure 4 CHO cell protection assay, which is one of the most important readout, the positive control vaccine from GSK performed better. Thus, is unclear if the two vaccines tested are worth development.
Response : Thank you for the valuable comment. We revised result section for 3.2 PTx CHO cell neutralization assay to clarify the results. GC DTaP showed equivalent results compared to commercial vaccine group and raP also did at pre infection that we assumed it is worth to develop in the future.
- Many figures are hard to read either not properly labeled or visually hard to read
Response : We changed the control group color to distinguish groups more readily and changed the y axis label for Fig 4 and Fig7 and resized the entire Figures.
- The writing is understandable but need extensive editing. For instance, line 229-230 "IgG and IgG1 showed almost identical patterns, which is believed to be because both subtypes show a Th2 orientation" is not clear. There are many other writings like this in this paper.
Response : Thank you for the valuable comment. We had major revision for clear conclusion and better quality of English. We also corrected line 229-230 for clear meaning Please refer to revised manuscript line 244-246.
Reviewer 2 Report
Comments and Suggestions for Authors
This manuscript describes the analysis of two new acellular pertussis based vaccines (aP) in a number of assays. The first vaccine is a traditional subunit based aP vaccine, similar to GSK’s Infanrix, and manufactured in Korea (GC DTaP). The second uses recombinant pertussis toxin subunit 1 (raP) instead of chemically detoxified PT. In mouse immunogenicity the raP generated a very poor response in terms of anti-PT IgG. GC DTaP and Infanrix were similar. For anti-FHA and Prn the new vaccines gave a look response. Surprisingly sera from mice immunised with raP gave a strong PT neutralising response in the CHO cell clustering assay. Mice immunised with raP also cleared B. pertussis following intranasal challenge at a slower rate than Infanrix and GC DTaP (the latter having the quicker rate of clearance). However, the raP immunised mice gave strong IFN, IL-17 and IL-5, and CD4 Trm responses.
With some revision I feel this paper is worth publishing despite the mixed results for raP. The authors have demonstrated that GC DTaP may be equivalent to licensed DTaP and therefore worth study in clinical trials. The promising results from raP shows it may be worth pursuing. Overall, the manuscript, especially the Abstract and Introduction, could be better written. Bacteria (eg E. coli and B. pertussis) should be in Italics.
Some points were confusing (see below) and need to be addressed.
Line 67 – Opening line should read “These outbreaks are probably due…”
Line 98 – the quantity of each antigen in Infanrix should be given or it should be stated that GC DTaP has the same quantities as Infanrix.
Line 104 – Is Subunit 1 mutated in anyway or has it been expressed in its native form?
Lines 217 – 226 This information is more suitable for the Discussion
Line 270 – is the CHO cell protection assay really just an assay for the presence of opsonizing antibodies in the test? Surely there are a range of antibodies that could prevent PT toxicity on CHO cells.
Line 350 -352 – the mock vaccine had a higher response 2 weeks post infection but at 1 week post infection raP had.
Line 353 – 354 – the vaccine groups don’t look equivalent in the figures and there are significant differences between some groups. This sentence needs to be clarified.
Line 356 – 357 – this line should be broken into 2 distinct sentences as deal with two sets of results
Lines 378-382 – is this more suited to the Discussion?
Line 403, 405 – should these be Fig 3?
Line 410 – the text of the CHO results says that the raP
Line 447 – but the difference wasn’t statistically significant
Line 473 – should be Figure 6b
Discussion - It would be useful to discuss any advantages and limitations associated with only using PT subunit 1 instead of the whole molecule
Comments on the Quality of English Language
The paper is understandable but some editing required.
Author Response
We would like to thank you for allowing us to revise the manuscript. We have revised the manuscript based on the comments. Additionally, we have made extensive revisions that we believe improved the quality of the manuscript. All authors have reviewed the revised version and approved it. Our point-by-point responses to the specific comments made by you and descriptions of additional changes are set out below.
Reviewer 2
- Overall, the manuscript, especially the Abstract and Introduction, could be better written.
Response : We appreciate your valuable comment. We revised the Abstract and Introduction section and also changed the title of the article.
- Bacteria should be in italics
Response : We would like to make a polite apology about the mistake about the italic issue. We corrected all the microbial scientific name in italics.
- line 67 : Opening line should read “These outbreaks are probably due”
Response : Thank you for the valuable comment. We revised the manuscript as advised. (line 63)
- line 98 : the quantity of each antigen in Infanrix should be given or it should be stated that GC DTaP had the same quantities as Infanrix.
Response : We appreciate your valuable comment. In the revised manuscript, more explanation about the components of InfanrixTM IPV/Hib vaccine and GC DTaP vaccine is inserted. InfanrixTM IPV/Hib vaccine contains 25 Lf diphtheria toxoid (DT), 10 Lf tetanus toxoid (TT), 25 μg PT, 25 μg FHA 25 μg and 8 μg PRN per 0.5mL. (line 102-108)
- line 104 : Is subunit 1 mutated in anyway or has it been expressed in its native form?
Response : The subunit 1 contains the enzymatically active site which is responsible for pathogenic consequence of pertussis toxin. In this study, a genetically detoxified S1 subunit (R9K/E129G) was used.
- line 217 - 226 : This information is more suitable for the Discussion
Response : We appreciate your valuable comment. We discussed the points you mentioned with authors thoroughly. We decided to leave the content as is in order to focus on the conclusion in the Discussion section. We ask for your deep understanding of this point.
- line 270 : Is the CHO cell protection assay really just an assay for the presence of opsonizing antibodies in the test? Surely there are a range of antibodies that could prevent PT toxicity on CHO cells
Response : Yes, The PTx neutralizing antibody (PTNAs) assay is one of the assay that quantify the opsonizing antibodies to PTx. We corrected the sentence “PTx CHO cell protection assay” to “PTx CHO cell neutralization assay” to clarify the meaning. We believed that the antibody range can be confirmed by comparing the neutralization titer to the ELISA standard under the assumption that there is a correlation between the ELISA result and the neutralization titer. However, in this study, a commercial vaccine was used as the standard, and there was no correlation between In house ELISA result and the neutralization titer.
- line 350-352 : the mock vaccine had a higher response 2 weeks post infection but at 1 week post infection raP had
Response : We appreciate your valuable comment. In the revised manuscript, we added “2 weeks post infection” instead “after infection”.
(line 351-353)
- line 353-354 : the vaccine groups don’t look equivalent in the figures and there are significant differences between some groups. This sentence needs to be clarified
Response : We appreciate your valuable comment. At 1 week after infection, the raP group had the highest secretion of IFN-gamma and IL-17a, so the contents were modified. (line 353-355)
- line 356-357 : this line should be broken into 2 distinct sentences as deal with two sets of results
Response : We appreciate your valuable comment. We corrected the whole paragraph (3.5 Cytokine Response) specially IL-5 secretion part for clear meaning. (line 355-360)
- line 378-382 : is this more suited to the Discussion?
Response : We appreciate your valuable comment and agree with it. We summarized the explanation and moved to the discussion section. (line 430-432)
- line 403-405 should there be Fig3?
Response : We would like to make a polite apology about the typing mistake. We corrected “Fig4” to “Fig 3”
- line 410 : the text of the CHO results says that the raP
Response : Thank you for valuable comment. Yes, the text explains with respect to raP group. We corrected the sentence properly please refer to revised manuscript line 419-420.
- line 447: but the difference wasn’t statistically significant
Response : We appreciate your valuable comment. For more accurate expression we revised the “The raP vaccine showed higher expression of CD4+ TRM cells compared to the aP vaccines.” sentence to “It should be noted that, in the case of the recombinant aP vaccine, the expression rate of CD4+ TRM cells was relatively higher than the other test groups (Figure 7b) at 2w p.i..” (line 440-441)
- Line 473 – should be Figure 6b
Response : We would like to make apology about the typing mistake. The true figure is FigS4 (supplementary Figure 4) and we corrected. Please refer to revised manuscript line 460.
- Discussion- It would be useful to discuss any advantages and limitations associated with only using PT subunit 1 instead of whole molecule
Response : The pertussis exotoxin is a multimeric, 105 kDa protein composed of 5 different subunits where S1 subunit carries an enzyme function that eventually interferes with G protein coupled signal pathways leading into pathogentic consequences by infection. The S1 subunit therefore provides a precise target for immunological intervention as vaccine antigen.
Reviewer 3 Report
Comments and Suggestions for Authors
The background of the paper is that the small-scale whooping cough has continued in Korea. Two DTaP vaccines have been developing in Korea. The authors evaluated the immunogenicities and bacterial challenges, and the results were compared to commercially available trivalent DTaP vaccines to evaluate the effectiveness of the developing aP vaccines. The results reaffirmed that antibodies to PT and neutralization antibodies are important for initial bacterial clearance. The paper is organized well. I suggest accepting the manuscript for publication.
Author Response
We would like to deeply thank you for giving us your valuable and positive comments about our manuscript. We have made revisions for more qualified manuscript but kept original meaning, please refer to a revised manuscript. Thank you again about your support.
Reviewer 4 Report
Comments and Suggestions for Authors
1. The authors recommend giving more attentional to the previous trials to create the pertussis vaccine (PT, FHA, PRA) and the recombination in their introduction such as, but not limited to PMID: 16734625, 9180182.
2. Line 99 “which contains 25 μg of PT, FHA) is this ug for both or for each one. It should be clear.
3. In line 100 after purification this component is absorbed on aluminum hydroxide and then combined with DT vaccine, while in line 110 the recombinant version is absorbed on Alhydrogel® adjuvant, is there are structural and/or adjuvanticity difference between Alhydrogel® adjuvant versus normal aluminum hydroxide?
4. Lines 103-109 data is the cornerstone in the manuscript, these data need more clarification otherwise need Reference(S).
5. Mal85 is a protein (domain) part coming from E. coli. if yes is this acceptable and/or safe?
6. Is Mal85 technology used for all raP vaccine molecules “PT, FHA, PRN”? Please clarify the important issue. if yes .. do you have this domain repeated three times? Is this correct?
7. As your vaccine version is recombinant and produced in E. coli, so please let us see the SDS-PAGE as well as the corresponding Western blot for each or combined raP molecules.
8. Please write a microbial scientific name in italics throughout the MS. Such as in line 109.
Round 2
Reviewer 1 Report
Comments and Suggestions for Authors
The quality of the manuscript has significantly increased in this version. All my questions were addressed.
Author Response
We would like to thank you deeply for your valuable comments that could be improve our manuscript. Thank you again for your rime and evaluation.
Reviewer 4 Report
Comments and Suggestions for Authors
Thank you very much.
Author Response

(The authors gave the same response as above.)
